# A numerical framework for optimal trade-offs between land use and LCOE using efficient, blockage-aware multi-fidelity methods

Cory Frontin, Jeffrey Allen, Christopher J. Bay, Jared Thomas, Ethan Young, and Pietro Bortolotti National Renewable Energy Laboratory, 15013 Denver West Parkway, Golden, CO 80401, USA **Correspondence:** Cory Frontin (cory.frontin@nrel.gov)

Abstract. This paper introduces a novel approach to efficiently estimate the annual energy production (AEP) of a wind farm. The numerical predictions are generated thanks to a multi-fidelity model that combines a classical low-fidelity wake engineering solver with a mid-fidelity computational fluid dynamic solver. The novel setup is not only faster than conventional approaches but is also capable of estimating the AEP of tightly spaced wind farms. Using this approach, we explore the trade-

5 off between land use and levelized cost of energy (LCOE) for a wind farm made of 25 turbines. The results of this study, which ignore impact the layout sensitivity of fatigue loads and their incumbent effect on costs, quantify the penalty on LCOE performance that can be paid to restrict the land use of a wind farm. The results also exhibit the novel capabilities of our approach for multi-fidelity wind farm design to avoid false optima according to incomplete representations of the relevant physical phenomena.

# 10 1 Background and motivation

The most prominent benefits of wind energy as a source of electricity are that it is inexhaustible, inexpensive, and secure. However, wind energy may also cause unwanted negative effects such as visual impact, shadow flicker, acoustic emissions, and mortality of wildlife. These negative effects not only impact communities and wildlife near wind farms but also harm the wind energy industry as a whole because they may delay, disrupt, and diminish the success of wind energy projects across

project life cycles.

Veers et al. (2023) highlighted the need for holistic design approaches to incorporate social and environmental metrics into the design of the next generation of wind energy systems. A major challenge to achieving this goal is that social and environmental factors are usually harder to pose as straightforward optimization targets. In this work, we take a first step in this direction by studying the trade-off between the levelized cost of energy (LCOE) and land use.

- We elect to study land use because it represents a simple and immediate metric that interfaces with both social and environmental factors. On the social side, land use (on a per-turbine or per-capacity basis) dictates the likelihood that a wind farm creates an unwanted negative impact. We work under the assumption that by minimizing the use of land, a wind farm minimizes its impact on neighbors. The assumption is supported by setback ordinances, which are used by government bodies to limit the proximity of wind turbines to structures or critical infrastructure. Setback ordinances significantly affect the deployment
- potential of wind energy (Lopez et al., 2023).

On the environmental side, the correlation between land use and impact on wildlife is harder to establish. Emerging research has investigated the impact of wind turbine designs and wind farm layouts on wildlife, specifically on raptors (Quon et al., 2022), bats (Hein and Straw, 2021), and sage grouse (LeBeau et al., 2017). In this work we assume that land use is correlated to the likelihood that a farm intersects with migratory pathways or habitats that put wildlife at risk of disruption or injury.

In terms of costs, reducing the land use of wind farms has a direct impact on energy losses and balance-of-system (BOS) costs. These two metrics tend to drive layout optimization in different directions: To minimize energy losses, which are caused by turbine wakes, turbines in a wind farm should be spaced as far apart as possible. Conversely, to minimize BOS costs, turbines should be installed closer to each other in a wind farm, as tighter spacing limits costs associated with new roads and cabling.

- In this work, we optimize wind farm layouts for LCOE, the most common metric for understanding the balance between the production of energy and the lifetime costs associated with energy production. This is not the first work in this direction. Fleming et al. (2016) developed a method for optimizing farm layout and controls with respect to expected farm power density or power given cable length constraints, but stopped short of full cost modeling. Other work considered trade-offs among noise, production, and land use (Yamani Douzi Sorkhabi et al., 2016) but without fully considering the physical impacts of the land use
- constraint. As the spacing of wind farms becomes tighter, technical challenges in the estimation of farm performance increase. Power estimates from engineering wake models are known to be inaccurate when turbines are in close proximity (Göçmen et al., 2016), though progress has been made in developing wake models with more accurate shape (Keane, 2021) and deficit (Zhang et al., 2023) in the near wake. These advancements can help improve wind farm performance estimation.
- Unfortunately, wake effects are the leading loss mechanism in wind farms, but they are not the only ones. Blockage effects must also be considered. The precise definition of blockage varies slightly in the literature, but the term refers to the deflection of airflow around an obstruction due to the pressures exerted by wind turbines and farms on the surrounding flow (Strickland and Stevens, 2022). Blockage effects can occur both upstream and downstream of the obstruction and can result in speedups and deficits in the flow. According to Sanchez Gomez et al. (2023), blockage effects are estimated to reduce the wind speed by 1 % to 5 %, and sometimes as much as 10 %, at a distance of 1.5 to 3 rotor diameters upstream of the first row of wind

turbines. While the focus of blockage study in wind energy is typically the negative upstream effects that slow the airflow into the front row of wind turbines, ignoring the downstream speedups and energy redistributing effects of blockage could also bias AEP predictions (Meyer Forsting et al., 2023). Wind farm blockage effects are typically neglected in engineering design tools, though progress is being made in the development of engineering design tools for estimating blockage effects. Nonetheless, all known state-of-the-art engineering-level blockage models neglect some effects of blockage and slowdown (Nygaard et al., 2020).

2020).

Blockage effects are caused by the interactions between the velocity and pressure fields, which cause incoming flow to stagnate at the front of a farm, reducing power yield. These effects fundamentally require flow-resolving computational fluid dynamics (CFD) estimates to capture accurate power estimates. While engineering tools seek to decouple the wake and blockage effects, these phenomena are inherently coupled. The combination of wake and blockage effects in a wind farm can be

captured more effectively through CFD than through engineering wake models; however, CFD approaches are much more computationally intensive than engineering models.

To balance the benefits and costs of both CFD and engineering model approaches, this paper applies a novel numerical framework to minimize the land usage for a structured five-by-five wind farm while efficiently incorporating engineering and CFD simulations to estimate annual energy production (AEP). In turn, LCOE estimates can be made more efficiently than

65 would be possible with CFD alone and more accurately than is possible with current engineering models. We show Pareto fronts describing the possible trade-offs in terms of land usage, power generation, and costs for a family of structured farms. These studies can inform both the policy decisions made by key stakeholder institutions and the design procedures adopted by wind farm developers, resulting in the maximization of wind energy deployment while limiting the impact on communities and wildlife.

### 70 2 Numerical framework

# 2.1 Optimization framework

The toolchain described in this manuscript aims to optimize the layout of wind farms by minimizing LCOE. LCOE is defined as the ratio of lifetime-averaged annual costs (C) and lifetime-averaged AEP and can be expressed in terms of U.S. dollars (USD) per megawatt-hour:

$$LCOE = \frac{C}{AEP} = \frac{r_{fc} \times (CapEx + BOS) + OpEx}{AEP}.$$
 (1)

In this equation, C is composed of operational expenditures (OpEx), including land lease costs, maintenance costs, and other recurring operational expenses; initial capital costs (CapEx), or the capital expenditures necessary to acquire the wind turbines and related equipment; and the BOS costs, including permits, transportation, roads, foundations, electrical connections, assembly, and installation. The overall capital costs, including the BOS costs, are multiplied by the fixed charge rate ( $r_{\rm fc}$ ) to determine their annualized cost;  $r_{\rm fc}$  represents the annual amount per dollar of the capital costs needed to cover all project

financing, including taxes, insurance, depreciation, return on debt and equity, and related fees<sup>1</sup>.
In Fig. 1, the information flow for the optimization process is shown in an extended design structure matrix (XDSM) diagram (Lambe and Martins, 2012). Each of the components, indicated by red rectangles, is interfaced and controlled by the Open-

- MDAO library (Gray et al., 2019). Flow solvers used in this work, the National Renewable Energy Laboratory's (NREL's)
  framework FLOw Redirection and Induction in Steady State (FLORIS) and NREL's finite-element solver WindSE, are described in Sect. 2.2.1 and 2.2.2, respectively. The multi-fidelity AEP integrator used in this work is developed in Sect. 2.3, and the land use calculator implements straightforward area calculations for a parallelogram-shaped farm. In this work, CapEx and OpEx are modeled as constant inputs. The BOS and LCOE calculators are from the Wind-Plant Integrated System Design and Engineering Model (WISDEM®), a Python framework also built around the OpenMDAO library. The wind farm parameteri-
- zation used to create the wind farm configuration design variables is described in Sect. 3.1. In this project, only some elements

<sup>&</sup>lt;sup>1</sup>If a project is realized by internal financing, then these monthly contributions can represent the equivalent internal costs of capital.