# Peer review of "A numerical framework for optimal trade-offs between land use and LCOE using efficient, blockage-aware multi-fidelity methods"

_Wind Energy Science, 2025_

## Author Comment (AC1)

**Summary of responses:**

| Commentor | Comment | Summarized Comment                                                                                                                                                                                      | Response                                                                                                                                                                                                                                                                                                                                                            |
|-----------|---------|---------------------------------------------------------------------------------------------------------------------------------------------------------------------------------------------------------|---------------------------------------------------------------------------------------------------------------------------------------------------------------------------------------------------------------------------------------------------------------------------------------------------------------------------------------------------------------------|
| AR #1     | 1.1     | Research scope is poorly defined, lacks focus between land-use considerations and multi-fidelity methods.                                                                                               | We have significantly re-structured the entire manuscript, to adhere to a clean IM-RaD format; additionally, the framing of the work has been shifted to concentrate on the multi-fidelity AEP method first with the LCOE/land use problem an example of an analysis enabled by and dependent on its use.                                                           |
|           | 1.2     | Lack of structure in the text, unnecessary repetition.                                                                                                                                                  | See response 1.1 to AR #1.                                                                                                                                                                                                                                                                                                                                          |
|           | 2       | Use of emphatic terms inflates impact and importance.                                                                                                                                                   | We have iterated the vocabulary and wording across the manuscript. Terms such as "significantly", "greatly", etc. have been removed entirely, and care has been taken to guarantee exaggerated and flowery language has been attenuated.                                                                                                                            |
|           | 3.1     | L17: "social and environmental factors are usually harder to pose as straightforward optimization targets" statement is not coherent because land-use is easy to constrain.                             | The land use/LCOE example has been reduced in scope generally, and we've highlighted how we use the land use/LCOE problem as a proxy example for analytically and quantitatively more complicated social and environmental factors. We've also improved our discussion on the relationship between turbine spacing and blockage and social outcomes more generally. |
|           | 3.2     | L21: provide references for negative land use effects; also land use does not necessarily lead to negative outcomes (e.g. financial compensation to landowners).                                        | We have added contextual discussion about the positive impacts of land use and the non-exclusivity of benefits in the small spacing direction.                                                                                                                                                                                                                      |
|           | 3.3     | Why use an LCOE minimization vs. AEP maximization? Most of the impacts should be anticipated to be due to increasing AEP.                                                                               | Our election to use the LCOE/land use problem because it is the relevant trade to high-level stakeholders, which is enabled by the technical work we've developed in this work. It also highlights the multidisciplinary analyses enabled by the multifidelity AEP estimation developed in this work.                                                               |
|           | 3.4     | L317: different fidelity models are only necessary iff there are a large differences in DVs across the different fidelities.                                                                            | We have addressed the language directly to point to the fact that the design variables on the FLORIS and multi-fidelity Pareto sets vary, and highlight in a few areas how design variable impacts drive the changes, including addressing comment 2.8 from AR #2.                                                                                                  |
|           | 3.5     | Grid-based parametrization limits the impact of the work for farm design problems, can't disentangle the effects between parameterization and model fidelity; generalization would be very interesting. | This is out of scope for this work but has been addressed in the restructured discussion section, under future work.                                                                                                                                                                                                                                                |

**Summary of responses (cont'd):**

| Commentor | Comment | Summarized Comment                                                                                                                                                                          | Response                                                                                                                                                                                                                                                                                                                                                                                                 |
|-----------|---------|---------------------------------------------------------------------------------------------------------------------------------------------------------------------------------------------|----------------------------------------------------------------------------------------------------------------------------------------------------------------------------------------------------------------------------------------------------------------------------------------------------------------------------------------------------------------------------------------------------------|
| AR #1     | 3.6     | L362-363: computational efficiency claims are not documented in the manuscript.                                                                                                             | We have re-framed the relevant section in the results to describe the computational savings in more detail, and we removed overly broad language about computational efficiency that are not strictly justified by results in the manuscript.                                                                                                                                                            |
|           | 4.1     | Use model (RANS, engineering wake model, etc.) names rather than NREL tool names.                                                                                                           | Any references to the specific (NREL-developed) tools have been removed except for in sections devoted to their descriptions or in subordinate clauses that include references to the types of models.                                                                                                                                                                                                   |
|           | 4.2     | XDSM useful but hard to read due to the use of symbols.                                                                                                                                     | The XDSM has been edited for clarity and additional commentary has been added per comment 2.1 of AR #2.                                                                                                                                                                                                                                                                                                  |
|           | 4.3     | Eq. (5) introduced, but all presented results appear to address Eq. (6).                                                                                                                    | We have re-worked the presentation of the optimization work generally, and removed the extraneous equations.                                                                                                                                                                                                                                                                                             |
|           | 4.4     | Some symbols (A and A_limit in Eq. (6)) are not introduced.                                                                                                                                 | This has been fixed in the re-working of the text.                                                                                                                                                                                                                                                                                                                                                       |
|           | 4.5     | No description or analysis of data in Fig. (9), difficult to follow the reasoning.                                                                                                          | Figures 9 and 10 have been consolidated and the resulting figure has been addressed comprehensively in the re-structured results section.                                                                                                                                                                                                                                                                |
|           | 4.6     | S3.2: results lack perspective with the literature, and comparison between low-fidelity, high-fidelity, and multi-fidelity models in terms of accuracy and computational effort is missing. | We enhanced the discussion of the direct comparison of the multi-fidelity method to the RANS-AD method for AEP computation in the text. Relevant validation work and error analysis across wind conditions with respect to high-cost AEP calculations was out of scope for this work due to project time and computational budget, but a relevant discussion has been placed in the future work section. |
|           | 5.1     | Repetitions, e.g. blockage effect def L44 and L56, MF GP at L284 & L285.                                                                                                                    | We have removed the duplications in the re-working of the manuscript; after restructuring, we've assessed for repitition.                                                                                                                                                                                                                                                                                |
|           | 5.2     | Unclear sentences: L286 "The standard deviation surfaces", L290 "At the lowest level,", L302 "At this point,".                                                                              | We clarified how the GP uses the standard deviation surfaces to calibrate the correction models in the multi-fidelity fit process in the text.                                                                                                                                                                                                                                                           |
|           | 5.3     | L329: "optimization termination points" not standard, use optimum or "solution of the optimization".                                                                                        | We have removed the non-standard terminology in the re-working of the text.                                                                                                                                                                                                                                                                                                                              |
|           | 5.4     | L324: "LCOE sub-optimalities" is not right, the points are optimal w.r.t. the model.                                                                                                        | We have removed the non-standard terminology in the re-working of the text.                                                                                                                                                                                                                                                                                                                              |

**Summary of responses (cont'd):**

| Commentor | Comment | Summarized Comment                                                                                                                                         | Response                                                                                                                                                                                                                                                                                                             |
|-----------|---------|------------------------------------------------------------------------------------------------------------------------------------------------------------|----------------------------------------------------------------------------------------------------------------------------------------------------------------------------------------------------------------------------------------------------------------------------------------------------------------------|
| AR #2     | 1.1     | Presentation of methodology is not clear.                                                                                                                  | With the re-structuring of the manuscript, we have concentrated on making the description of the methodology more clear, including a structured description of the multi-fidelity scheme, aerodynamic solvers, multi-disciplinary integration, and optimization problems.                                            |
|           | 1.2     | Optimization framework given, but connections between steps is not clearly stated; details of the method and optimization procedure not provided in depth. | We have significantly re-structured the entire manuscript, to adhere to a clean IM-RaD format; additionally, the framing of the work has been shifted to concentrate on the multi-fidelity AEP method first with the LCOE/land use problem an example of an analysis enabled by and dependent on its use.            |
|           | 1.3     | Computational setup not presented clearly.                                                                                                                 | See response 1.1 to AR #2.                                                                                                                                                                                                                                                                                           |
|           | 1.4     | Application of land use optimization is weakly presented but results contain contradicting statements.                                                     | The framing of the work has been shifted to concentrate on the multi-fidelity AEP method first with the LCOE/land use problem an example of an analysis enabled by and dependent on its use.                                                                                                                         |
|           | 2.1     | S2.1, basic steps of the procedure should be given (starting point, etc.) along with XDSM.                                                                 | This has been addressed with comprehensive changes in the text where the XDSM figure is mentioned.                                                                                                                                                                                                                   |
|           | 2.2     | L159-L167: computational domain set up are not clear, incl. total domain size in X,Y,Z, and not related to key lengths such as D.                          | We have improved the presentation of the computational details to be consistent with where and how key dimensional quantities are introduced elsewhere in the paper. Our x- and y-ranges were [-4550, 4550] m and our z-range was [0, 825] m.                                                                        |
|           | 2.3     | Z-direction size not given, and doesn't allow assessment of refinement; refinement near the ground is unclear vs turbine region refinement.                | We have introduced dimensional quantities for the z-range in the presentation of the computational details, see response 2.2 to AR #2. Additionally, we have made clearer how the nested grid refinements lead to cells with dimensional length scales in the proximity of the rotor plane.                          |
|           | 2.4     | " four levels of nested localized refinements surrounding each turbine" is unclear; maybe add a sketch.                                                    | The language used in presenting the specifics of the nested refinement levels has been improved and separated for clarity. Additionally, we have attempted to clarify how the nesting process reduces the length scales, and how we used those scaling rules to achieve the necessary refinement in the rotor plane. |

**Summary of responses (cont'd):**

| Commentor | Comment | Summarized Comment                                                                                                                    | Response                                                                                                                                                                                                                        |
|-----------|---------|---------------------------------------------------------------------------------------------------------------------------------------|---------------------------------------------------------------------------------------------------------------------------------------------------------------------------------------------------------------------------------|
| AR #2     | 2.5     | L169-L171: "upstream/downstream non-horizontal boundaries" too wordy and imply more than intended, just name them.                    | We have improved the description of the aerodynamic solvers along with the restructuring into the IMRaD formatting.                                                                                                             |
|           | 2.6     | Fig. (3) L2 is supposed to be parallel w.r.t. the side of the domain; also helpful if approx. flow direction is shown on this sketch. | We have made changes to the layout diagram and adjoining text to resolve ambiguities.                                                                                                                                           |
|           | 2.7     | L322: LCOE v. L333-334 LCOE statements are contradictory (HF LCOE higher than LF LCOE vs. MF LCOE is less than the LF LCOE).          | We have re-composed and re-organized the text in the optimization results, and made changes in how the descriptions of the LCOE estimates are given in order to increase the clarity about the results in Fig. 9 (formerly 10). |
|           | 2.8     | L335-336: lengths given additionally in terms of rotor diameters in addition to square kilometers.                                    | This has been adjusted in the text by the addition of parenthetical nondimensional quantities.                                                                                                                                  |
|           | 2.9     | S4.2: necessity of multi-start optimization is stressed but neglected earlier in the manuscript.                                      | Handling of the multi-start capability has been disambiguated by an extended description in the re-structured optimization section and an accompanying table.                                                                   |